# What matters when building vision-language models?

**Hugo Laurençon**[*,1,2]    **Léo Tronchon**[*,1]    **Matthieu Cord**[2,3]    **Victor Sanh**[1]

[1]Hugging Face [2]Sorbonne Université [3]valeo.ai, Paris, France
[*]The order was chosen randomly.

## Abstract

The growing interest in vision-language models (VLMs) has been driven by improvements in large language models and vision transformers. Despite the abundance of literature on this subject, we observe that critical decisions regarding the design of VLMs are often not justified. We argue that these unsupported decisions impede progress in the field by making it difficult to identify which choices improve model performance. To address this issue, we conduct extensive experiments around pre-trained models, architecture choice, data, and training methods. Our consolidation of findings includes the development of Idefics2, an efficient foundational VLM of 8 billion parameters. Idefics2 achieves state-of-the-art performance within its size category across various multimodal benchmarks, and is often on par with models four times its size. We release the model (base, instructed, and chat) along with the datasets created for its training.

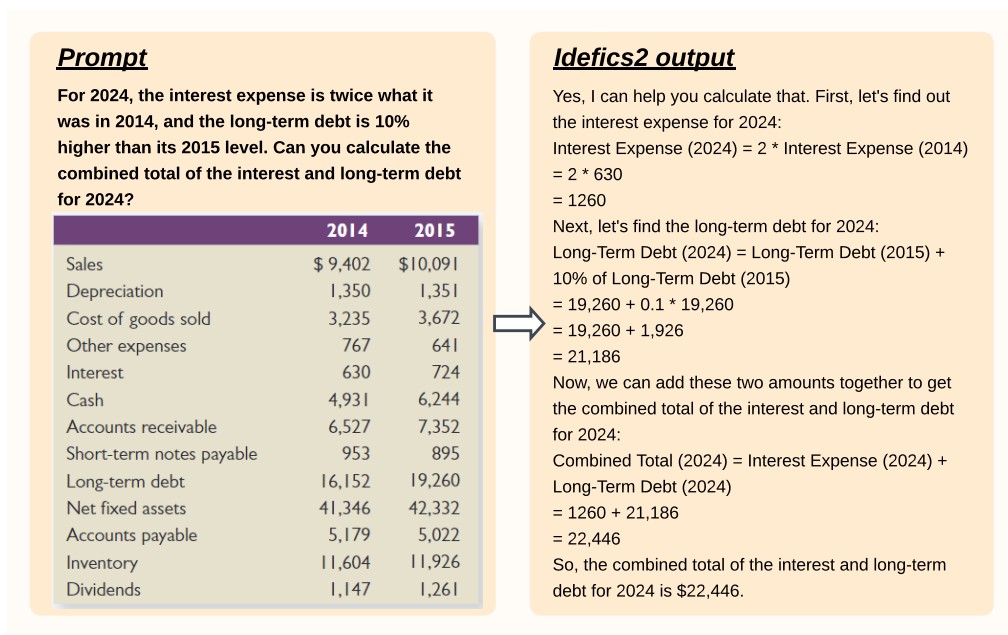

Figure 1: Idefics2-chatty analyzes the table to compute and answer the query.

38th Conference on Neural Information Processing Systems (NeurIPS 2024).

# 1 Introduction

Vision-language models (VLMs) that take images and texts as inputs and output texts, are useful for many tasks, like retrieving information in a scanned PDF (Hu et al., 2024), explaining charts or diagrams (Carbune et al., 2024), transcribing the text in an image (Blecher et al., 2023), counting objects in a picture (Goyal et al., 2017) or turning screenshots of webpages into code (Laurençon et al., 2024). The development of powerful open large language models (Touvron et al., 2023; Jiang et al., 2023; Team et al., 2024) and image encoders (Zhai et al., 2023; Sun et al., 2023; Radford et al., 2021) enables researchers to build upon these unimodal pre-trained models to create advanced VLMs that solve these problems with increasing accuracy (Dai et al., 2023; Liu et al., 2023; Bai et al., 2023; Lin et al., 2023; Li et al., 2023; Wang et al., 2023). Despite the progress in the field, the literature reveals many disparate design choices which are often not justified experimentally, or very briefly.

This situation makes it challenging to distinguish which decisions truly account for model performance, thereby making it difficult for the community to make meaningful and grounded progress. For instance, (Alayrac et al., 2022; Laurençon et al., 2023) use interleaved Transformer-based cross-attentions to fuse the image information into the language model, while (Li et al., 2023; Liu et al., 2023) concatenate the sequence of image hidden states with the sequence of text embeddings, and feed the concatenated sequence to the language model. To our knowledge, this choice has not been properly ablated, and trade-offs in terms of compute, data efficiency and performance are poorly understood. In this work, we aim to bring experimental clarity to some of these core design choices and pose the question: **What matters when building vision-language models?**

We identify two areas where various works adopt different design choices: (a) model architecture, and in particular, connector modules that fuse the vision and text modalities and their impact on inference efficiency, (b) multimodal training procedure and its impact on training stability. For each of these areas, we rigorously compare different design choices in a controlled environment and extract experimental findings. Notably, we find that (a) the progress of vision-language models is in large part driven by the progress of pre-trained unimodal backbones, (b) the more recent fully autoregressive architecture outperforms the cross-attention architecture, although it requires modifications to the optimization procedure to ensure a stable training, (c) adaptation of the pre-trained vision backbone and the modules connecting the text and vision modalities allow for more efficiency at inference time on one side, and handling images in their original ratio and size without harming downstream performance on the other side, and (d) modifications to the image processing enables trading inference cost for downstream performance.

Our results are complementary with those presented in (Karamcheti et al., 2024; McKinzie et al., 2024; Lin et al., 2023) which derive insights about multi-stage training, selective unfreezing of the pre-trained backbones, data repetition, and impact of training mixture on zero and few-shot performance. We specifically analyze unexplored aspects such as model architecture, training methods, stability, and efficiency improvements at inference. For example, we introduce the task of text transcription from PDFs directly in the pre-training, we justify the benefits of using synthetic captions in image-text pair datasets, we detail our multi-stage training procedure to save computational resources, and we create a large-scale instruction fine-tuning dataset.

Learning from these insights, we train Idefics2, a foundational VLM with 8 billion parameters. Idefics2 achieves state-of-the-art performance in its size category on various benchmarks while being more efficient at inference, for both the base and the fine-tuned version. It is on par with state-of-the-art models 4 times larger on some vision-language benchmarks and matches the performance of Gemini 1.5 Pro on some challenging benchmarks. We release the base, instructed, and chat versions of Idefics2[1] as resources for the VLM community along with the data created to train the model.

# 2 Terminology

We first establish shared terminology for discussing the different design choices. Training VLMs typically requires gluing together a pre-trained vision backbone and a pre-trained language backbone by initializing new parameters to connect the two modalities. Training these new parameters is done during the *pre-training phase*. This stage commonly leverages a large multimodal dataset such as image-caption pairs. We note that even though it is most common to start from two separate

---

[1]https://huggingface.co/collections/HuggingFaceM4/idefics2-661d1971b7c50831dd3ce0fe

unimodal pre-trained backbones, the parameters of these two backbones can be optionally shared and initialized from scratch as done in (Bavishi et al., 2023). As in the large language models literature, the pre-training stage is followed by an instruction fine-tuning stage, in which the model learns from task-oriented samples.

Recent works explore two main choices to combine the visual inputs and the text inputs. In the *cross-attention architecture* (Alayrac et al., 2022; Laurençon et al., 2023; Awadalla et al., 2023), the images encoded through the vision backbone are injected at different layers within the language model by interleaving cross-attention blocks in which the text cross-attends to the image hidden states. In contrast, in the *fully autoregressive architecture* (Koh et al., 2023; Driess et al., 2023; Liu et al., 2023), the output of the vision encoder is directly concatenated to the sequence of text embeddings, and the entire sequence is passed as input to the language model. The input sequence of the language model is thus the concatenation of *visual tokens* and text tokens. The sequence of visual tokens can be optionally pooled into a shorter sequence, providing more compute efficiency. We refer to the layers that maps the vision hidden space to the text hidden space as *modality projection* layers. Figure 2 highlights the fully-autoregressive architecture we ultimately use for Idefics2.

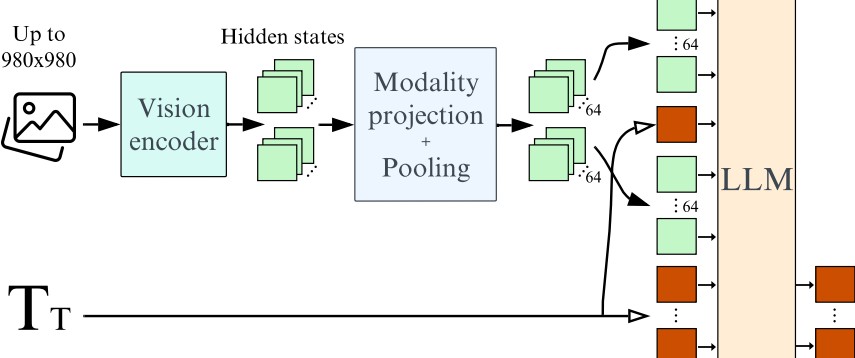

Figure 2: Idefics2 fully-autoregressive architecture: Input images are processed by the Vision encoder. The resulting visual features are mapped (and optionally pooled) to the $LLM$ input space to get the visual tokens ($64$ in our standard configuration). They are concatenated (and potentially interleaved) with the input sequence of text embeddings (green and red column). The concatenated sequence is fed to the language model ($LLM$), which predicts the text tokens output.

## 3   Exploring the design space of vision-language models

In this section, we compare recurrent design choices in the vision-language model literature and highlight findings. Unless specified otherwise, we run the ablations for 6'000 steps and report the average score of the 4-shot performance on 4 downstream benchmarks measuring different capabilities: VQAv2 (Goyal et al., 2017) for general visual question answering, TextVQA (Singh et al., 2019) for OCR abilities, OKVQA (Marino et al., 2019) for external knowledge, and COCO (Lin et al., 2014) for captioning. We run the ablations on eight nodes containing eight H100s each, for up to five days.

### 3.1   Are all pre-trained backbones equivalent for VLMs?

Most recent VLMs start from pre-trained unimodal backbones. How does the choice of the backbones (vision and text) influence the performance of the resulting VLM?

We fix the size of the pretrained backbones, the data used for multimodal pre-training, and the number of training updates. Under the cross-attention architecture, we observe that the greatest improvement in the performance on vision-language benchmarks comes from changing the language model to a better one. More specifically, replacing LLaMA-1-7B (Touvron et al., 2023) (35.1% on MMLU (Hendrycks et al., 2021)) by Mistral-7B (Jiang et al., 2023)

| LM backbone | Avg. score |
|---|---|
| Llama-1-7B | 62.5 |
| Mistral-7B | 67.6 |

Table 1: Ablation on the language model backbone.

(60.1% on MMLU) yields a boost of 5.1 (see Table 1). Additionally, switching the vision encoder from CLIP-ViT-H (Radford et al., 2021) (78.0% on ImageNet(Deng et al., 2009)) to SigLIP-SO400M (Zhai et al., 2023) (83.2% on ImageNet) yields a 3.3 increase in performance on the benchmarks (see Table 2). This result on better vision backbones corroborates observations from (Karamcheti et al., 2024).

We note that Chen and Wang (2022) reports a stronger increase in performance by scaling the size of the vision encoder compared to scaling the size of the language model even though scaling the vision encoder leads to a smaller parameter count increase. Although EVA-CLIP-5B (Sun et al., 2023) is ten times bigger in parameter counts than SigLIP-SO400M (Zhai et al., 2023), we obtain similar performance across 4 benchmarks, suggesting that EVA-CLIP-5B could be heavily under-trained,

| VE backbone | Res. | Avg. score |
|---|---|---|
| CLIP-ViT-H | 224 | 57.4 |
| EVA-CLIP-5B | 224 | 60.2 |
| SigLIP-SO400M | 384 | 60.7 |

Table 2: Ablation on the vision encoder backbone.

and we acknowledge that the open VLM community is missing a large well-trained vision encoder.

> ***Finding* 1.** Better pre-trained LLMs and vision encoders lead to better performance on multimodal tasks. However, with the best current models for both, the LLM has a higher impact.

### 3.2 How does the fully autoregressive architecture compare to the cross-attention architecture?

To our knowledge, there is no proper comparison between the fully autoregressive and the cross-attention architecture. We aim to fill this gap by considering their trade-offs, namely performance, parameter count, and inference cost.

Following (Alayrac et al., 2022), we first compare the two architectures by freezing the uni-modal backbones and training only the newly initialized parameters (cross-attention on one side, and modality projection along with learned pooling on the other side), while fixing the amount of training data.

| Architecture | Backbones training | # params | # trainable params | Avg. score |
|---|---|---|---|---|
| Fully autoreg. no Perceiver | Frozen | 7.6B | 5M | 51.8 |
| Fully autoreg. | Frozen | 8.3B | 740M | 60.3 |
| Cross-attention | Frozen | 10B | 2.3B | 66.7 |
| Cross-attention | LoRA | 10B | 2.5B | 67.3 |
| Fully autoreg. | LoRA | 8.3B | 950M | 69.5 |

Table 3: Ablation for the architecture and method of training.

Alayrac et al. (2022) shows that the more frequently the cross-attention blocks are interleaved with the language model layers, and the higher the vision-language performance. As such, we note that under this setup, the cross-attention architecture has 1.6B more trainable parameters (2.3B trainable parameters in total) than the fully autoregressive architecture. Additionally, at inference time, the former uses 10% more flops than the latter. Under these conditions, we observe that the cross-attention architecture performs 7 points better in Table 3.

Out of the total number of parameters, approximately 10% for the fully autoregressive architecture and 25% for the cross-attention are trained. We hypothesize that this low proportion limits the expressivity of the training and hinders performance. To test that hypothesis, we compare the two architectures by unfreezing all parameters (newly initialized parameters and parameters of the pre-trained unimodal backbones). Under these conditions, training the fully autoregressive architecture would yield loss divergences, and we were not successful in stabilizing the training even by aggressively lowering the learning rate or gradually unfreezing various components. To overcome this stability challenge, we leverage Low-Rank Adaptation (Hu et al., 2022) to adapt the pre-trained parameters while using standard full fine-tuning for the newly initialized ones.

This setup yields significantly more stable trainings, and more importantly, we observe a 12.9 points increase under the fully autoregressive architecture, and 0.6 point under the cross-attention architecture. While the cross-attention architecture performs better than the fully autoregressive architecture with frozen backbones, it is worse when we add degrees of liberty for the pre-trained backbones. Besides, using LoRA allows training the unimodal backbones at a fraction of the GPU memory cost of full fine-tuning, and LoRA layers can be merged back into the original linear layers

yielding no additional cost at inference. We therefore choose the fully autoregressive architecture in the rest of this work.

It is interesting to note that this finding contradicts (Karamcheti et al., 2024) in which the authors observed that unfreezing the pre-trained visual backbone would significantly degrade the performance. We hypothesize that using parameter-efficient fine-tuning methods is a key difference.

> ***Finding 2.*** The cross-attention architecture performs better than the fully autoregressive one when unimodal pre-trained backbones are kept frozen. However, when training the unimodal backbones, the fully autoregressive architecture outperforms the cross-attention one, even though the latter has more parameters.

> ***Finding 3.*** Unfreezing the pre-trained backbones under the fully autoregressive architecture can lead to training divergences. Leveraging LoRA still adds expressivity to the training and stabilizes it.

### 3.3 Where are the efficiency gains?

**Number of visual tokens** Recent VLMs typically route the entire sequence of the vision encoder's hidden states directly into the modality projection layer, which subsequently inputs into the language model, with no pooling. This is motivated by previous works in which adding a pooling strategy, like average pooling, was found to deteriorate the performance (Vallaeys et al., 2024). This results in a high number of visual tokens for each image ranging from 576 for DeepSeek-VL (Lu et al., 2024) to 2890 for SPHINX-2k (Lin et al., 2023). With the resulting sequence lengths, training is computationally costly, and in-context learning with interleaved images and texts is challenging because it requires modifications to the language models to handle very large context windows.

We reduce the sequence length of each image's hidden states by using a perceiver resampler (Jaegle et al., 2021) as a form of trainable Transformer-based pooling. The number of queries (also referred to as latents) corresponds to the number of resulting visual tokens after the pooling (from one for classification Touvron et al. (2021) to several in Alayrac et al. (2022) or a growing number for continual learning Douillard et al. (2022)). We observe that the learned pooling is effective in two ways: it increases the performance by 8.5 points on average and reduces the number of visual tokens necessary for each image from 729 to 64 (see Table 3).

In contrast to (Vallaeys et al., 2024; McKinzie et al., 2024) which find that the more visual tokens the higher the performance, we observe no gains when using more than 64 visual tokens. Other variations over the Perceiver architecture (Mañas et al., 2023; Darcet et al., 2024; Vallaeys et al., 2024) resulted in decreased performance.

| Pooling | # vis. tok. | Avg. score |
|---------|-------------|------------|
| Perceiver | 128 | 71.2 |
| Perceiver | 64 | 71.7 |

Table 4: Ablation on the pooling strategy.

> ***Finding 4.*** Reducing the number of visual tokens with learned pooling significantly improves compute efficiency at training and inference while improving performance on non-OCR downstream tasks.

**Preserving the original aspect ratio and image resolution** Vision encoders, such as SigLIP, are typically trained on fixed-size square images. Resizing images alters their original aspect ratio, which is problematic, for instance, for tasks requiring reading long texts. Furthermore, conditioning the training on a single resolution size inherently introduces limitations: a low resolution omits crucial visual details, while a high resolution leads to inefficiency in training and inference. Allowing the model to encode images at various resolutions allows users to decide how much compute is spent on each image.

Following Lee et al. (2023); Dehghani et al. (2023), we pass the image patches to the vision encoder without resizing the image or modifying its aspect ratio. Given that SigLIP was trained on fixed-size low-resolution square images, we interpolate the pre-trained positional embeddings to allow for

| Images | Res. | Avg. score |
|--------|------|------------|
| Square images | 768 | 73.1 |
| AR preserving | 378-768 | 72.1 |

Table 5: Ablation on the aspect-ratio preserving strategy.

a higher resolution and train the vision encoder with LoRA
parameters to adapt to these modifications.[2]
Our findings indicate that the aspect ratio preserving strategy maintains performance levels on downstream tasks while unlocking computational flexibility during both training and inference (see Table 5). In particular, not having to resize images to the same high resolution allows for saving GPU memory and handling images at the resolution they require.

> ***Finding 5.*** Adapting a vision encoder pre-trained on fixed-size square images to preserve images' original aspect ratio and resolution does not degrade performance while speeding up training and inference and reducing memory.

### 3.4 How can one trade compute for performance?

(Lin et al., 2023; Li et al., 2023; Liu et al., 2024; McKinzie et al., 2024) show that splitting an image into sub-images allows boosting the downstream performance with no changes to the model's signature. An image is decomposed into sub-images (for instance 4 equal sub-images), which are then concatenated to the original image to form a sequence of 5 images. Additionally, the sub-images are resized to the original image's size. This strategy however comes at the cost of a much higher number of tokens to encode the images.

We adopt this strategy during the instruction fine-tuning stage. Each single image becomes a list of 5 images: 4 crops and the original image. This way, at inference, the model is able to deal with standalone images (64 visual tokens per image), as well as artificially augmented images (320 visual tokens in total per image). We notice that this strategy is particularly useful for benchmarks like TextVQA and DocVQA, which require a sufficiently high resolution to extract the text in an image (see Table 9).

Moreover, when we apply image splitting to only 50% of the training samples (instead of 100% of the samples), we observe that it does not impair the performance increase that image splitting provides. Surprisingly, we find at evaluation time that increasing the resolution of the sub-images (and the standalone image) provides only a minor boost in performance compared to the improvement yielded by sole image splitting: 73.6% when increasing the resolution of the sub-images to the maximum vs 73.0% accuracy on our validation set of TextVQA, and respectively 72.7 vs 72.9 ANLS on the validation set of DocVQA.

> ***Finding 6.*** Splitting images into sub-images during training allow trading compute efficiency for more performance during inference. The increase in performance is particularly noticeable in tasks involving reading text in an image.

## 4 Idefics2 - an open state-of-the-art vision-language foundation model

With these learnings in hand, we train an open 8B parameters vision-language model: Idefics2. This section describes the construction of the model, the choice of the dataset, the sequence of training phases and compares the resulting model against VLMs baselines.

### 4.1 Multi-stage pre-training

We start from SigLIP-SO400M and Mistral-7B-v0.1 and pre-train Idefics2 on 3 types of data.

**Interleaved image-text documents** We use OBELICS (Laurençon et al., 2023), an open web-scale dataset of interleaved image-text documents with 350 million images and 115 billion text tokens. As shown by the authors, the long documents of OBELICS allow for preserving the performance of the language model while learning to deal with an arbitrary number of interleaved images and texts and long context. Additionally, the authors show that interleaved image-text documents are the biggest driving factor in increasing the performance on visual question answering (VQA) tasks, in particular

---

[2]Since SigLIP is trained with a fixed resolution, the positional embeddings can be interpreted both as absolute or relative positions. With the aspect ratio and resolution preserving, these positions become relative positional embeddings.

in the in-context learning setup. We perform an additional removal of newly opted-out content in January 2024 using the Spawning API[3] even though OBELICS had already been filtered to exclude opted-out content as of September 2023. We also removed the 5% of documents with the highest perplexity scores, as computed by Falcon-1B (Penedo et al., 2023).

**Image-text pairs**  Training on image-text pairs allows the model to learn the alignment between images and their associated texts. We use a combination of high-quality human-annotated image-text pairs from PMD (Singh et al., 2022) and higher-noise web-scale image-text pairs from (Schuhmann et al., 2022). To limit the amount of poor-quality data, we opt for the synthetic captions obtained through the LAION COCO[4] version of the dataset where images have been captioned with a model trained on COCO. This improves the quality of the training samples and

| Captions | Avg. score |
|---|---|
| Alt-texts | 49.8 |
| Synthetic | 52.9 |

Table 6: Ablation on synthetic captions against alt-text for image-text pairs.

thus the quality of the resulting model (see Table 6). We use a NSFW classifier[5] with a high recall and remove 7% of the samples in LAION COCO. We manually inspect 5'000 examples and found 28 pornographic images in the original LAION COCO and only 1 after filtering. This filtering does not negatively impact the downstream performance.

**PDF documents**  Sun et al. (2023) shows that a large proportion of mistakes of state-of-the art VLMs stem from their failure to accurately extract text in images or documents. In order to obtain strong OCR and document understanding abilities, we train Idefics2 on different sources of PDF documents: 19 million industry documents from OCR-IDL (Biten et al., 2022) and 18 million pages from PDFA[6]. Moreover, we add Rendered Text[7] to complement the dataset with texts written with a wide variety of fonts and colors and on diverse backgrounds. These integrations significantly boost the per-

| OCR data | Res. | DocVQA |
|---|---|---|
| W/o | 384 | 22.6 |
| W/o | 768 | 42.9 |
| W/ | 768 | 49.9 |

Table 7: Ablation on the synergy between OCR data and image resolution. We pre-trained the models for 5'500 steps, followed by 500 steps of fine-tuning on DocVQA.

formance on benchmarks that require reading text without decreasing the performance on other benchmarks (see Table 7).

To maximize compute efficiency, we decompose the pre-training in two stages. In the first stage, we limit the max image resolution to 384 pixels, which allows us to use a large global batch size of 2'048 (17k images and 2.5M text tokens on average). We sample OBELICS for 70% of the examples with a maximum sequence length of 2'048, and the image-text pairs datasets for 30% of the examples with a maximum sequence length of 1'536. In the second stage, we introduce PDF documents. Since they require a higher image resolution for the text to be legible, we increase the resolution to a maximum of 980 pixels. We use the same global batch size, but have to decrease the per-device batch size and use gradient accumulation to compensate for the additional memory cost. OBELICS represents 45% of the examples with a maximum sequence length of 2'048, image-text pairs represent 35% of the examples with a maximum sequence length of 1'536, and PDF documents represent the remaining 20% of the examples with a maximum sequence length of 1'024. Additionally, we randomly scale up images to adequately cover the distribution of potential image sizes. We emphasize that the training stages are different than the ones ablated in (Karamcheti et al., 2024): instead of selectively freezing/unfreezing parts of the model, we train the entire model during both stages (some parameters are trained with LoRA) and increase the image resolution from one stage to the other.

We use a learning rate of $10^{-4}$ with AdamW for the optimizer, and do around 2 epochs on our training data. It corresponds to approximately 1.5 billion images and 225 billion text tokens. We note that this is orders of magnitude more training data than other open VLMs. For example, ShareGPT (Chen et al., 2023) uses 1.2 million images, while Monkey (Li et al., 2023) uses 1.4 million for training. In total, we use 32 nodes of eight H100s each for 3 weeks for the multi-stage pre-training.

To evaluate the base model, we consider VQAv2 (Goyal et al., 2017), TextVQA (Singh et al., 2019), OKVQA (Marino et al., 2019), and COCO (Lin et al., 2014). Table 8 presents the results. While having fewer tokens per image, and thus being more efficient, Idefics2-base performs favorably

---

[3]https://spawning.ai/
[4]https://laion.ai/blog/laion-coco/
[5]https://github.com/LAION-AI/LAION-SAFETY
[6]https://huggingface.co/datasets/pixparse/pdfa-eng-wds
[7]https://huggingface.co/datasets/wendlerc/RenderedText

| Model | Size | Archi. | # tokens per image | VQAv2 | TextVQA | OKVQA | COCO |
|---|---|---|---|---|---|---|---|
| OpenFlamingo | 9B | CA | - | 54.8 | 29.1 | 41.1 | 96.3 |
| Idefics1 | 9B | CA | - | 56.4 | 27.5 | 47.7 | 97.0 |
| Flamingo | 9B | CA | - | 58.0 | 33.6 | 50.0 | 99.0 |
| MM1 | 7B | FA | 144 | 63.6 | 46.3 | 51.4 | **116.3** |
| Idefics2-base | 8B | FA | **64** | **70.3** | **57.9** | **54.6** | 116.0 |

Table 8: Performance of Idefics2-base against state-of-the-art base VLMs. The evaluations were done with 8 random in-context examples, and in an open-ended setting for VQA tasks.
*FA: fully autoregressive architecture. CA: cross-attention architecture.*
*(Task, Metric, Split): (VQAv2, VQA acc., testdev), (TextVQA, VQA acc., val), (OKVQA, VQA acc., val), (COCO, CIDEr, test)*

compared to the other current best base VLMs (OpenFlamingo (Awadalla et al., 2023), Idefics1 (Laurençon et al., 2023), Flamingo (Alayrac et al., 2022), and MM1 (McKinzie et al., 2024)). It is notably much better at reading texts in an image. Figure 3 shows an example of an output from the base model on a task similar to the pre-training.

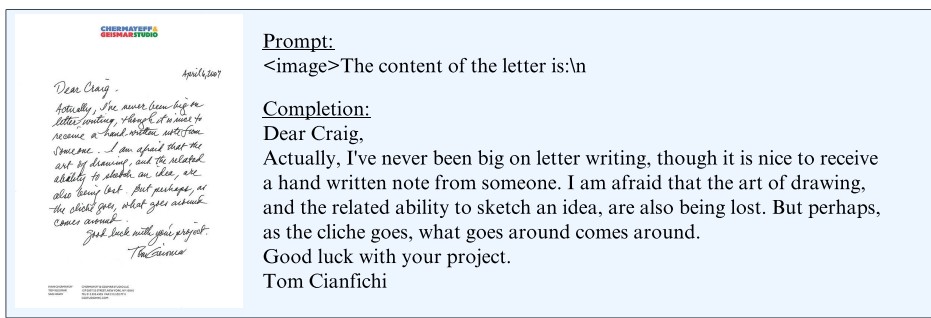

Figure 3: An example of text transcription with Idefics2-base.

## 4.2 Instruction fine-tuning

We continue the training with an instruction fine-tuning phase.

To do so, we create and release The Cauldron[8], a massive collection of 50 vision-language datasets, covering a wide range of tasks: general visual question answering, counting, captioning, text transcription, document understanding, chart/figure understanding, table understanding, visual reasoning, geometry, spotting differences between 2 images or converting a screenshot to a functional code. Similarly to (Sanh et al., 2022; Wei et al., 2022; Bach et al., 2022; Dai et al., 2023; Li et al., 2023), each dataset is prompted into a shared question/answer format. When there are multiple question/answer pairs per image, we concatenate the pairs into a multi-turn conversation. We deduplicate the training set against the evaluation sets, ensuring that there is minimum contamination from the training to the evaluation.

In addition to these vision-language datasets and following insights from (McKinzie et al., 2024), we add text-only instruction datasets to the mixture. The datasets aim at teaching the model to follow complex instructions, solve mathematical problems, or do arithmetic calculations. We give more details about the chosen datasets, the number of images, question-answer pairs, and size of each of the subsets, as well as our selected mixture proportion in Table 14 in Appendix A.2.1.

We instruction-tune the base model using DoRA (Liu et al., 2024) (a variant of LoRA). During the fine-tuning, we only compute the loss on the tokens of the answers in the Q/A pairs. Since we are doing many epochs over some of the datasets, we employ several strategies to lower the risk of overfitting. First, we add noise to the embeddings with the NEFTune (Jain et al., 2024) technique.

---

[8]https://huggingface.co/datasets/HuggingFaceM4/the_cauldron

| Model | Size | # tokens per image | MMMU | MathVista | TextVQA | MMBench |
|---|---|---|---|---|---|---|
| LLaVA-NeXT | 13B | 2880 | 36.2/- | 35.3 | 67.1 | 70.0 |
| DeepSeek-VL | 7B | 576 | 36.6/- | 36.1 | 64.4 | 73.2 |
| MM1-Chat | 7B | 720 | 37.0/35.6 | 35.9 | 72.8 | 72.3 |
| Idefics2 | 8B | **64** | **43.5/37.9** | **51.6** | 70.4 | **76.8** |
| Idefics2 | 8B | 320 | 43.0/37.7 | 51.4 | **73.0** | 76.7 |

Table 9: Performance of Idefics2 against state-of-the-art VLMs up to a size of 14B parameters. The evaluations are done in zero shot. Idefics2 with 64 or 320 tokens per image is the same model (same weights), only the inference differs. The full table is present in Appendix A.3.2.
*(Benchmark, Split, Metric): (MMMU, val/test, MMMU score), (MathVista, testmini, MMMU score), (TextVQA, val, VQA acc.), (MMBench, test, accuracy).*

Then, we scale up randomly the resolution of the images during the training. Finally, when applicable, we shuffle the multiple user/assistant turns randomly before feeding the example to the model.

We evaluate Idefics2 on commonly adopted benchmarks: MMMU (Yue et al., 2024) for multidiscipline college-level problems, MathVista (Lu et al., 2024) for mathematical reasoning, TextVQA (Singh et al., 2019) for text reading on natural images, and MMBench Liu et al. (2023) for various perception and reasoning tasks. Table 9 presents the results (see Table 15 for the complete result table) of Idefics2 against the current strongest VLMs in its class size: LLaVA-Next (Liu et al., 2024), DeepSeek-VL (Lu et al., 2024) and MM1-Chat (McKinzie et al., 2024). While being computationally much more efficient at inference, Idefics2 exhibits strong performance on various benchmarks, outperforming the current best foundation VLMs in its size category. It is on par with state-of-the-art models 4x its size, or with closed-source models like Gemini 1.5 Pro on several benchmarks like MathVista or TextVQA. The detailed performance of Idefics2 across each category of MMMU (Yue et al., 2024) is present in Table 16.

### 4.3 Optimizing for chat scenarios

The evaluation benchmarks expect very short answers, but humans prefer long generations when interacting with a model. We find that Idefics2 can exhibit difficulties in precisely following instructions about the expected format, making it difficult to reconcile "chattiness" and downstream performance. As such, after instruction fine-tuning, we further train Idefics2 on dialogue data. We fine-tune Idefics2 for a few hundred steps on LLaVA-Conv (Liu et al., 2023) and ShareGPT4V (Chen et al., 2023), with a large batch size. Our blind human evaluations reveal that Idefics2-chatty is overwhelmingly preferred over its instruction fine-tuned version in many user interactions. We also adversarially stress-tested the model to generate inaccurate, biased, or offensive responses and reported the findings in Appendix A.5. We show examples of generations with Idefics2-chatty in Figure 1, and in Appendix in Figures 5, 6 and 7.

## 5 Conclusion

In this work, we re-examine common choices made in the VLM literature and rigorously compare these choices in controlled experiments. Our findings touch upon the effectiveness of different architectures, their performance/inference cost trade-offs as well as training stability. With these learnings at hand, we train Idefics2, an open 8B parameters vision-language model. Idefics2 is state-of-the-art on various benchmarks in its category size and is much more efficient at inference. By releasing our findings, as well as our models and our training dataset, we aim to contribute to the ongoing evolution of VLMs and their applications in solving complex real-world problems.

## Acknowledgement

We thank Mustafa Shukor for helpful suggestions on the paper, and Yacine Jernite, Sasha Luccioni, Margaret Mitchell, Giada Pistilli, Lucie-Aimée Kaffee, and Jack Kumar for red-teaming the model.

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

# A  Appendix

## A.1  Further experimental details of the ablations

### A.1.1  Cross-attention vs. fully-autoregressive architectures

We apply LoRA modules to the LLM for the fully autoregressive architecture and to the cross-attention modules and the LLM for the cross-attention architecture. In Figure 4, we report the average performance with respect to the number of steps, the number of images, as well as the number of text tokens. We see an improvement across the board with the fully autoregressive architecture. Comparing the average score with these different axes is essential because the cross-attention architecture feeds a single token per image to the language model, against 64 for the fully autoregressive architecture with perceiver pooling. This implies that for the same training sequence length, the number of images and text tokens is different for the two architectures. Equivalently, the same multimodal document will yield different sequence lengths. Even though we fix the batch size in the comparison, the number of text tokens and number of images grow at different paces under the two architectures.

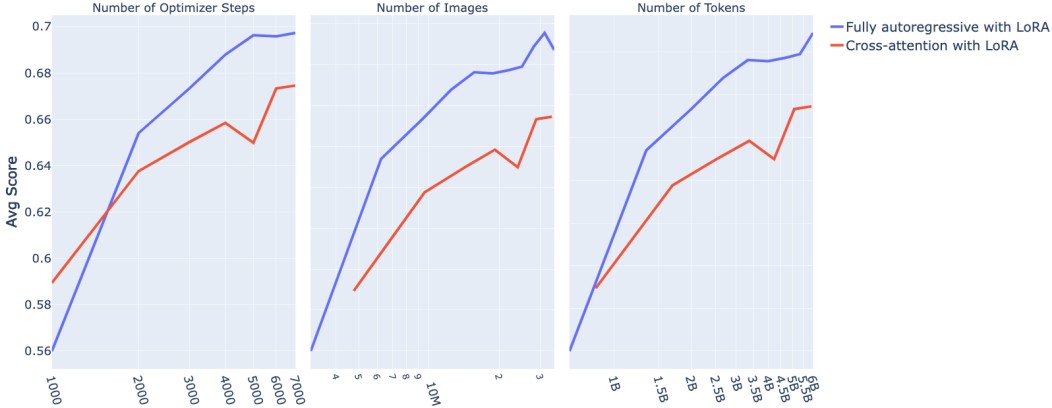

Figure 4: Comparison of the cross-attention and fully autoregressive architectures through the number of steps, the number of images and the number of text tokens.

### A.1.2  Comparing various vision backbones

We present in Table 10 the detailed results of comparing multiple vision backbones. While EVA-CLIP-5B performs similarly to SigLIP-SO400M, we emphasize that it has 11 times more parameters. We also noticed in early experiments that TextVQA is the most sensitive benchmark to image resolution, which accounts for the performance increase.

| VE backbone | Size | Res. | Avg. score | VQAv2 | OKVQA | TextVQA | COCO |
|---|---|---|---|---|---|---|---|
| CLIP-ViT-H | 600M | 224 | 57.4 | 52.4 | 41.7 | 28.2 | 107.5 |
| EVA-CLIP-5B | 4.4B | 224 | 60.2 | 53.4 | 43.3 | 30.4 | **113.7** |
| SigLIP-SO400M | 400M | 384 | **60.6** | **53.6** | **43.4** | **33.8** | 111.6 |

Table 10: Detailed results of ablation on the vision encoder backbone

### A.1.3  Comparing various pooling strategies

We compare multiple pooling strategies: a simple linear layer that takes the flattened sequence of vision hidden states and projects it into a shorter sequence of visual tokens, as well as a Mapping Network (Mañas et al., 2023). The perceiver resampler significantly outperforms these two options (see Table 11).

We also ablate the number of layers in the perceiver resampler, and find no statistically significant differences when increasing the number of layers, similarly to results from Xiao et al. (2024). We settle on 3 layers out of caution to avoid any potential capacity bottleneck.

Finally, we add a 2-layer modality projection MLP on top of the vision encoder hidden states to project the vision hidden dimension to the language model hidden dimension prior to the perceiver resampler. These changes yield better performance as well (see Table 13).

| Vision-language Connector | Avg. score |
|---|---|
| Linear Projection | 44.5 |
| Mapping Network (Mañas et al., 2023) | 51.8 |
| Perceiver | 60.3 |

Table 11: Ablation on the modality projection

| Num. perceiver layers | Avg. score |
|---|---|
| 1 | 69.3 |
| 3 | 68.6 |
| 12 | 69.0 |

Table 12: Ablation on the number of perceiver resampler layers

| MLP modality projection | Avg. score |
|---|---|
| W/ | 71.4 |
| W/o | 69.6 |

Table 13: Ablation on the addition of a modality projection before the perceiver resampler

### A.1.4 Ablations on OCR data

We hypothesize that adding PDF documents helps the model learn to read text from images. In Table 7, we compare checkpoints trained with and without OCR documents, along with image resolution increase to ensure that the text is legible. We do not observe statistically significant differences when evaluating checkpoints in zero or few shot. Instead, we fine-tune the checkpoints on DocVQA for 500 steps with a learning rate of $1e - 5$, leading to checkpoints showing much stronger differences.

### A.1.5 Statistical significance of the experiments

In the context of large-scale experiments, this paper does not include error bars due to the high costs and time demands of multiple runs; each experiment can take as long as five days on eight nodes containing eight H100s each. Nevertheless, we conducted multiple runs for some experiments and observed differences of less than one point in the average score across all cases. This minimal variation supports the validity of the conclusions drawn from our ablation studies.

## A.2 Details of the instruction fine-tuning

### A.2.1 Statistics of The Cauldron

In Table 14, we present the statistics of the datasets included in The Cauldron, as well as the text-only instruction datasets used for the supervised fine-tuning. For each dataset, we give the number of different images it contains, the number of question-answer pairs, the total number of tokens for the answers in the question-answer pairs, and the selected percentage of tokens it represents in our final mixture after upsampling or downsampling.

| Dataset | # images | # Q/A pairs | # tokens | % mixture |
|---|---|---|---|---|
| *General visual question answering* | | | | |
| VQAv2 (Goyal et al., 2017) | 82,772 | 443,757 | 1,595,929 | 5.72% |
| COCO-QA (Ren et al., 2015) | 46,287 | 78,736 | 286,982 | 1.47% |
| Visual7W (Zhu et al., 2016) | 14,366 | 69,817 | 279,268 | 1.43% |
| A-OKVQA (Schwenk et al., 2022) | 16,539 | 17,056 | 236,492 | 1.21% |
| TallyQA (Acharya et al., 2019) | 98,680 | 183,986 | 738,254 | 0.57% |
| OK-VQA (Marino et al., 2019) | 8,998 | 9,009 | 38,853 | 0.40% |
| HatefulMemes (Kiela et al., 2020) | 8,500 | 8,500 | 25,500 | 0.13% |
| VQA-RAD (Lau et al., 2018) | 313 | 1,793 | 8,418 | 0.09% |
| *Captioning* | | | | |
| LNarratives (Pont-Tuset et al., 2020) | 507,444 | 507,444 | 21,328,731 | 4.56% |

| | | | | |
|---|---|---|---|---|
| Screen2Words (Wang et al., 2021) | 15,730 | 15,743 | 143,103 | 0.37% |
| VSR (Liu et al., 2023) | 2,157 | 3,354 | 10,062 | 0.21% |
| | | | | |
| *OCR, document understanding, text transcription* | | | | |
| RenderedText[9] | 999,000 | 999,000 | 27,207,774 | 5.57% |
| DocVQA (Mathew et al., 2021) | 10,189 | 39,463 | 337,829 | 3.46% |
| TextCaps (Sidorov et al., 2020) | 21,953 | 21,953 | 389,658 | 2.00% |
| TextVQA (Singh et al., 2019) | 21,953 | 34,602 | 181,918 | 1.86% |
| ST-VQA (Biten et al., 2019) | 17,247 | 23,121 | 127,846 | 1.31% |
| OCR-VQA (Mishra et al., 2019) | 165,746 | 801,579 | 6,073,824 | 0.93% |
| VisualMRC (Tanaka et al., 2021) | 3,027 | 11,988 | 168,828 | 0.86% |
| IAM (Marti and Bunke, 2002) | 5,663 | 5,663 | 144,216 | 0.74% |
| InfoVQA (Mathew et al., 2022) | 2,118 | 10,074 | 61,048 | 0.63% |
| Diagram image-to-text[10] | 300 | 300 | 22,196 | 0.11% |
| | | | | |
| *Chart/figure understanding* | | | | |
| Chart2Text (Obeid and Hoque, 2020) | 26,985 | 30,242 | 2,852,827 | 4.38% |
| DVQA (Kafle et al., 2018) | 200,000 | 2,325,316 | 8,346,234 | 4.27% |
| VisText (Tang et al., 2023) | 7,057 | 9,969 | 1,245,485 | 1.91% |
| ChartQA (Masry et al., 2022) | 18,271 | 28,299 | 185,835 | 1.90% |
| PlotQA (Methani et al., 2020) | 157,070 | 20,249,479 | 8478299.278 | 0.65% |
| FigureQA (Kahou et al., 2017) | 100,000 | 1,327,368 | 3,982,104 | 0.61% |
| MapQA (Chang et al., 2022) | 37,417 | 483,416 | 6,470,485 | 0.33% |
| | | | | |
| *Table understanding* | | | | |
| TabMWP (Lu et al., 2023) | 22,729 | 23,059 | 1,948,166 | 2.49% |
| TAT-QA (Zhu et al., 2021) | 2,199 | 13,215 | 283,776 | 2.18% |
| HiTab (Cheng et al., 2022) | 2,500 | 7,782 | 351,299 | 1.80% |
| MultiHiertt (Zhao et al., 2022) | 7,619 | 7,830 | 267,615 | 1.37% |
| FinQA (Chen et al., 2021) | 5,276 | 6,251 | 242,561 | 0.99% |
| WikiSQL (Zhong et al., 2017) | 74,989 | 86,202 | 9,680,673 | 0.99% |
| SQA (Iyyer et al., 2017) | 8,514 | 34,141 | 1,894,824 | 0.97% |
| WTQ (Pasupat and Liang, 2015) | 38,246 | 44,096 | 6,677,013 | 0.51% |
| | | | | |
| *Reasoning, logic, maths* | | | | |
| GeomVerse (Kazemi et al., 2024) | 9,303 | 9,339 | 2,489,459 | 3.83% |
| CLEVR-Math (Lindström and al, 2022) | 70,000 | 788,650 | 3,184,656 | 3.26% |
| CLEVR (Johnson et al., 2017) | 70,000 | 699,989 | 2,396,781 | 1.23% |
| IconQA (Lu et al., 2021) | 27,315 | 29,859 | 112,969 | 1.16% |
| RAVEN (Zhang et al., 2019) | 42,000 | 42,000 | 105,081 | 0.67% |
| Inter-GPs (Lu et al., 2021) | 1,451 | 2,101 | 8,404 | 0.17% |
| | | | | |
| *Textbook/academic questions* | | | | |
| AI2D (Kembhavi et al., 2016) | 3,099 | 9,708 | 38,832 | 0.80% |
| TQA (Kembhavi et al., 2017) | 1,496 | 6,501 | 26,004 | 0.53% |
| ScienceQA (Lu et al., 2022) | 4,985 | 6,218 | 24,872 | 0.25% |
| | | | | |
| *Differences between 2 images* | | | | |
| NLVR2 (Suhr et al., 2019) | 50,426 | 86,373 | 259,119 | 1.33% |
| GSD (Li et al., 2023) | 70,939 | 141,869 | 4,637,229 | 0.48% |
| Spot the diff (Jhamtani et al., 2018) | 8,566 | 9,524 | 221,477 | 0.57% |
| | | | | |
| *Screenshot to code* | | | | |
| WebSight (Laurençon et al., 2024) | 500,000 | 500,000 | 276,743,299 | 0.28% |
| DaTikz (Belouadi et al., 2023) | 47,974 | 48,296 | 59,556,252 | 0.03% |

---

[9] https://huggingface.co/datasets/wendlerc/RenderedText

[10] https://huggingface.co/datasets/Kamizuru00/diagram_image_to_text

*Text-only general instructions, math problems, arithmetic calculations*

| | | | | |
|---|---|---|---|---|
| OpenHermes-2.5 (Teknium, 2023) | 0 | 1,006,223 | 248,553,747 | 12.73% |
| LIMA (Zhou et al., 2023) | 0 | 1,052 | 633,867 | 0.81% |
| Dolly (Conover et al., 2023) | 0 | 14,972 | 1,329,999 | 0.68% |
| MetaMathQA (Yu et al., 2024) | 0 | 395,000 | 74,328,255 | 3.81% |
| MathInstruct (Yue et al., 2024) | 0 | 261,781 | 45,393,559 | 2.33% |
| OrcaMath (Mitra et al., 2024) | 0 | 200,031 | 63,780,702 | 1.63% |
| CamelAIMath (Li et al., 2023) | 0 | 49,744 | 21,873,629 | 0.06% |
| AtlasMathSets[11] | 0 | 17,807,579 | 455,411,624 | 3.50% |
| Goat (Liu and Low, 2023) | 0 | 1,746,300 | 167,695,693 | 0.86% |

Table 14: The statistics of datasets used for instruction fine-tuning. # tokens is the total number of tokens for each dataset for the answers only. % mixture is our selected percentage of answer tokens for each dataset in the final mixture.

### A.3 Details of the evaluations

#### A.3.1 Evaluation setup

We perform all evaluations with a batch size of 1 and greedy decoding.

For the multi-choice questions in MMMU, MathVista, MMBench, we evaluate with the same prompt used for similar types of datasets during the instruction fine-tuning:

> Question: {question}
> Choices:
> A. {choice_a}
> B. {choice_b}
> C. {choice_c}
> ...
> Answer with the letter.

For the open-ended questions in TextVQA, DocVQA, and VQAv2, we evaluate with the prompt:

> Question: {question}
> Give a very brief answer.

We use the stop words `Question`, `User`, `<end_of_utterance>` and `<eos>` to stop a generation.

#### A.3.2 Expanded evaluation table

We report the expanded evaluation of Idefics2 and the comparison to other models in Table 15. This includes scores on VQAv2 (Goyal et al., 2017), which is widely adopted for evaluation. We acknowledge, though, that the metric used for the open-ended visual question answering benchmarks strongly penalizes models that do not generate in the same format as the ground truth. For example, answering "large" when the ground truth is "big" or more verbose reformulations will be counted as incorrect. Our manual qualitative analysis reveals that on benchmarks like VQAv2, the generations of two models differing by 5 points would be barely noticeable. This problem is less concerning for other open-ended benchmarks like TextVQA or DocVQA which require finding a text in an image, making the expected answer less prone to ambiguity. The detailed performance of Idefics2 across each category of MMMU (Yue et al., 2024) is present in Table 16.

#### A.3.3 Qualitative evaluation

We show in Figures 5, 6, and 7, examples of generations with Idefics2-chatty.

---

[11]https://huggingface.co/datasets/AtlasUnified/atlas-math-sets

| Model | Size | # tokens per image | MMMU | MathVista | TextVQA | MMBench | DocVQA | VQAv2 |
|---|---|---|---|---|---|---|---|---|
| *7B-14B models* | | | | | | | | |
| LLaVA-NeXT | 13B | 2880 | 36.2/- | 35.3 | 67.1 | 70.0 | - | 82.8 |
| DeepSeek-VL | 7B | 576 | 36.6/- | 36.1 | 64.4 | 73.2 | 49.6 | - |
| MM1-Chat | 7B | 720 | 37.0/35.6 | 35.9 | 72.8 | 72.3 | - | 82.8 |
| Idefics2 | 8B | 64 | 43.5/37.9 | 51.6 | 70.4 | 76.8 | 67.3 | 80.8 |
| Idefics2 | 8B | 320 | 43.0/37.7 | 51.4 | 73.0 | 76.7 | 74.0 | 81.2 |
| *≥30B models* | | | | | | | | |
| Mini-Gemini-HD | 34B | 2880 | 48.0/44.9 | 43.3 | 74.1 | 80.6 | - | - |
| MM1-Chat | 30B | 720 | 44.7/40.3 | 39.4 | 73.5 | 75.1 | - | 83.7 |
| LLaVA-NeXT | 34B | 2880 | 51.1/44.7 | 46.5 | 69.5 | 79.3 | - | 83.7 |
| *Proprietary* | | | | | | | | |
| Gemini 1.0 Pro | - | - | 47.9/- | 45.2 | 74.6 | - | 88.1 | 71.2 |
| Claude 3 Haiku | - | - | 50.2/- | 46.4 | - | - | 88.8 | - |
| Claude 3 Sonnet | - | - | 53.1/- | 47.9 | - | - | 89.5 | - |
| Gemini 1.5 Pro | - | - | 58.5/- | 52.1 | 73.5 | - | 86.5 | 73.2 |

Table 15: Performance of Idefics2 against state-of-the-art VLMs across different sizes. The evaluations are done in zero shot. Idefics2 with 64 or 320 tokens per image only differs by the image splitting.
*(Benchmark, Split, Metric): (MMMU, val/test, MMMU score), (MathVista, testmini/test, MMMU score), (TextVQA, val, VQA acc.), (MMBench, test, accuracy), (DocVQA, test, ANLS score), (VQAv2, testdev, VQA acc.).*

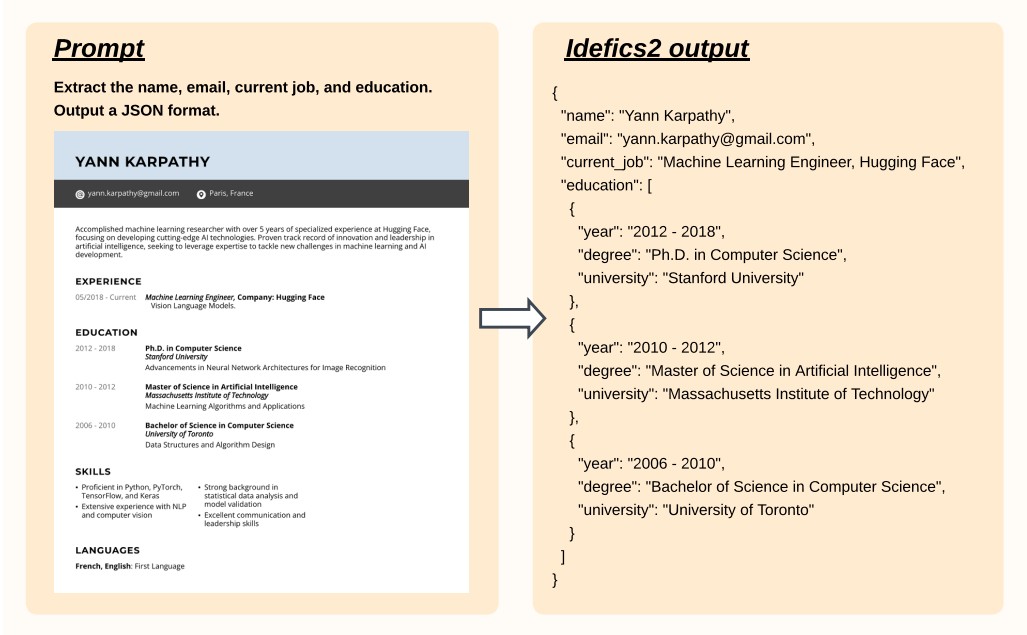

Figure 5: Idefics2-chatty finds the requested information in the resume, and organizes it in JSON format.

| MMMU category | Score |
|---|---|
| *Overall* | 45.3 |
| *Accounting* | 26.7 |
| *Agriculture* | 53.3 |
| *Architecture and Engineering* | 33.3 |
| *Art* | 76.7 |
| *Art Theory* | 76.7 |
| *Basic Medical Science* | 53.3 |
| *Biology* | 36.7 |
| *Chemistry* | 43.3 |
| *Clinical Medicine* | 43.3 |
| *Computer Science* | 43.3 |
| *Design* | 76.7 |
| *Diagnostics and Laboratory Medicine* | 36.7 |
| *Economics* | 46.7 |
| *Electronics* | 36.7 |
| *Energy and Power* | 23.3 |
| *Finance* | 40.0 |
| *Geography* | 30.0 |
| *History* | 50.0 |
| *Literature* | 83.3 |
| *Manage* | 46.7 |
| *Marketing* | 40.0 |
| *Materials* | 33.3 |
| *Math* | 56.7 |
| *Mechanical Engineering* | 43.3 |
| *Music* | 40.0 |
| *Pharmacy* | 36.7 |
| *Physics* | 26.7 |
| *Psychology* | 36.7 |
| *Public Health* | 36.7 |
| *Sociology* | 53.3 |
| *Art & Design* | 67.5 |
| *Business* | 40.0 |
| *Health & Medicine* | 41.3 |
| *Humanities & Social Science* | 55.8 |
| *Science* | 38.7 |
| *Tech & Engineering* | 38.1 |

Table 16: Detailed performance of Idefics2 across each category of MMMU (Yue et al., 2024).

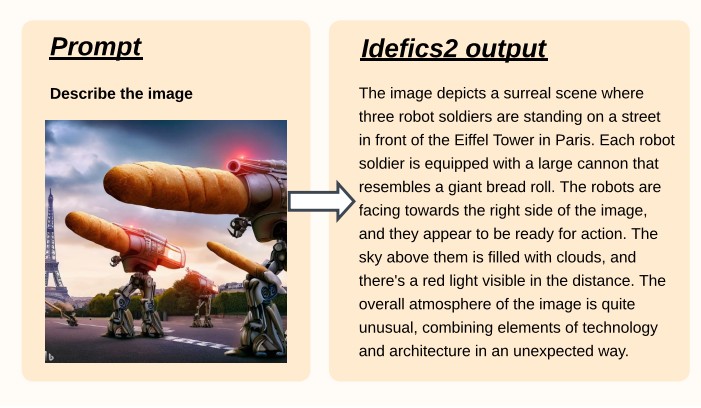

Figure 6: Idefics2-chatty describes an AI-generated image.

## A.4 Limitations

Our research highlights significant advancements in VLMs, yet several limitations persist.

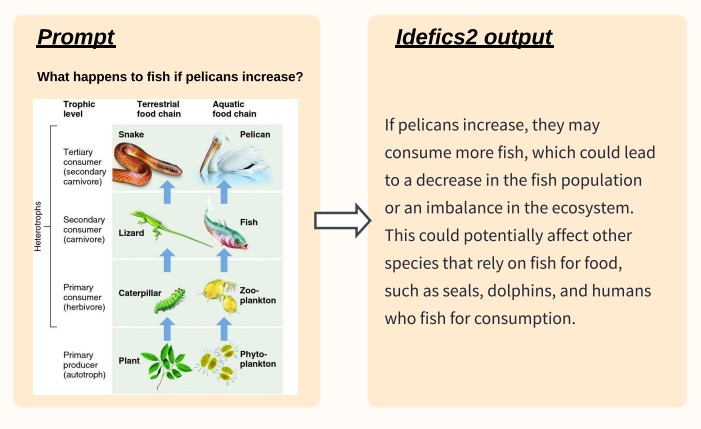

Figure 7: Idefics2-chatty answers a question on a scientific diagram.

Notably, current VLMs, including Idefics2, are prone to hallucinations, often generating content by describing nonexistent elements in images or fabricating facts. This issue underscores the challenges in ensuring the accuracy and reliability of model outputs.

Furthermore, the development of these models is slowed down by the lack of openly available, large-scale pre-trained vision encoders and comprehensive human-created instruction datasets with long answers for fine-tuning. Such resources are crucial for advancing VLMs, especially those optimized for conversational applications.

Additionally, as discussed in Section A.5, our models can be manipulated or "jailbroken" to produce outputs that are non-inclusive or disrespectful. This vulnerability highlights the importance of continuing to improve the robustness and ethical alignment of VLMs to prevent misuse and ensure they contribute positively in diverse applications.

## A.5  Red-teaming

In the context of a red-teaming exercise, our objective is to evaluate the propensity of the model to generate inaccurate, biased, or offensive responses. We evaluate more specifically the chat-optimized checkpoint[12].

While the model typically refrains from responding to offensive inputs, we observe that through repeated trials or guided interactions, it tends to hastily form judgments in situations necessitating nuanced contextual understanding, often perpetuating harmful stereotypes. Noteworthy instances include:

- Speculating or passing judgments, or perpetuating historical disparities on individuals' professions, social status, or insurance eligibility based solely on visual cues (e.g., age, attire, gender, facial expressions).
- Generating content that promotes online harassment or offensive memes reinforcing harmful associations from a portrait, or from a benign image.
- Assuming emotional states or mental conditions based on outward appearances.
- Evaluating individuals' attractiveness solely based on their visual appearance.

Additionally, we identify behaviors that increase security risks that already exist:

- Successfully solving CAPTCHAs featuring distorted text within images.
- Developing phishing schemes from screenshots of legitimate websites to deceive users into divulging their credentials.

---

[12]https://huggingface.co/HuggingFaceM4/idefics2-8b-chatty

- Crafting step-by-step guides on constructing small-scale explosives using readily available chemicals from common supermarkets or manipulating firearms to do maximum damage.

It's important to note that these security concerns are currently limited by the model's occasional inability to accurately read text within images.

We emphasize that the model would often encourage the user to exercise caution about the model's generation or flag how problematic the initial query can be in the first place. For instance, when insistently prompted to write a racist comment, the model would answer that query before pointing out "*This type of stereotyping and dehumanization has been used throughout history to justify discrimination and oppression against people of color. By making light of such a serious issue, this meme perpetuates harmful stereotypes and contributes to the ongoing struggle for racial equality and social justice.*".

However, certain formulations can circumvent (i.e. "jailbreak") these cautionary prompts, emphasizing the need for critical thinking and discretion when engaging with the model's outputs. While jail-breaking text LLMs is an active research area, jail-breaking vision-language models have recently emerged as a new challenge as vision-language models become more capable and prominent (Shayegani et al., 2024). The addition of the vision modality not only introduces new avenues for injecting malicious prompts but also raises questions about the interaction between vision and language vulnerabilities.

### A.6 Licenses

#### A.6.1 License for the models

The model is built on top of two pre-trained models: SigLIP-SO400M (Zhai et al., 2023) and Mistral-7B-v0.1 (Jiang et al., 2023). Both were released under the Apache 2.0 license, and we release the Idefics2 checkpoints under the same license.

#### A.6.2 License for the Cauldron

Each of the publicly available sub-datasets present in the Cauldron are governed by specific licensing conditions. Therefore, when making use of them, you must take into consideration each of the licenses governing each dataset. To the extent we have any rights in the prompts, these are licensed under CC-BY-4.0.

