# OpenReview forum: "What matters when building vision-language models?"
_NeurIPS.cc/2024/Conference — NeurIPS 2024 poster_

### Official Review · Reviewer_FVDr · 2024-06-26

**Soundness:** 4
**Presentation:** 4
**Contribution:** 4
**Rating:** 6
**Confidence:** 5

**Summary:**

This work conducts comprehensive experiments for re-examing common choices in the VLM area, such as unimodal backbone, connector, number of visual tokens, data, etc. Based on their experiments, they observe several findings important to the VLM research community. Finally, they rely on their key findings to collect corresponding training datasets and train a powerful VLM.

**Strengths:**

The writing of this work is clear and easy to follow, and the experiments are thoroughly conducted.  I have to say, I appreciate the thorough experiments and clear reasoning presented in this work. The findings obtained during the experiments can also contribute to the advancement of the VLM community.

**Weaknesses:**

Since the authors ultimately chose a fully autoregressive training approach, why did they still use cross-attention training for the ablation study in Section 3.1 to support finding 1? Is it because finding 1 does not hold under the autoregressive training approach?

**Questions:**

See weaknesses above.

---

> ### Author Rebuttal · Authors · 2024-08-03
>
> Thank you FVDr for your review. We appreciate your positive feedback regarding the strengths of our work.
>
> Regarding the point mentioned about section 3.1, we can clarify our approach:
>
> We wanted to start by showing some results with the cross-attention architecture that was the reference approach at that time. This is why the ablations in Section 3.1 were conducted under the cross-attention setup. However, we have verified that the two most significant findings (replacing Llama1 with Mistral and CLIP with SigLIP) also hold under the fully autoregressive approach. Given that these results were consistent with those obtained from the cross-attention architecture early in training, we did not extend these additional training to full completion to save computational resources. As a result, these ablations were not included in the final paper.
>
> We will be pleased to add these results with comments in the final paper for clarity and readability. We also ensure that our conclusions are in line with recent competing work in the literature.

---

> > ### Comment · Reviewer_FVDr · 2024-08-11
> > **Answer to the authors**
> >
> > Thanks for your response. I choose to keep my score unchanged.

---

### Official Review · Reviewer_vo36 · 2024-07-09

**Soundness:** 3
**Presentation:** 4
**Contribution:** 3
**Rating:** 7
**Confidence:** 4

**Summary:**

This paper empirically investigates several essential design choices of vision-language models, such as language backbones, visual encoder backbones, and different architectures. Through extensive experiments, they evaluate different model architectures, training methods, and data utilization strategies. Based on their findings, the authors trained a foundational VLM with 8B parameters that shows state-of-the-art performance.

**Strengths:**

One of the key strengths of this paper is its focus on an important and timely topic. By addressing the critical decisions in VLM design and highlighting areas where improvements can be made, this research contributes to a deeper understanding of how to build more effective and efficient models.

The analysis and conclusions presented in the paper are particularly insightful and have the potential to significantly benefit the AI research community. By identifying key factors that influence VLM performance, such as the choice of model architecture and training methods, the authors provide clear and actionable recommendations for building better models.

Besides, the availability of the trained models and public code is a notable strength of this work. Additionally, the paper is well-written and well-structured, making it easy to follow. Overall, this is a good paper, and I recommend accepting it.

**Weaknesses:**

The baseline models compared in the study are primarily open-source models. It would be beneficial to include and discuss state-of-the-art (SOTA) commercial models to provide a more comprehensive evaluation of the model's performance.

Additionally, incorporating performance reports of more open-source models, as listed on platforms like [MMBench](https://mmbench.opencompass.org.cn/leaderboard), in Tables 8 and 9 could better position the model's ability within the current research community. It is noticed that several open-source models of similar size achieve better performance. For instance, models like [InternLM-XComposer2-VL](https://github.com/InternLM/InternLM-XComposer) demonstrate superior performance despite having a comparable size, indicating potential areas for improvement in the proposed model.

The evaluation of the model's in-context learning ability is relatively limited. It would be useful to investigate whether the model's performance improves with an increased number of demonstrations, which could provide further insights into its capabilities and limitations in real-world applications.

**Questions:**

**Availability and Openness**

- Will the training code and datasets also be public for the community? Ensuring the availability of both the training code and datasets would greatly enhance the reproducibility of the study and allow the research community to build upon this work.

**Real-World Applicability**

- How well does the model handle real-world scenarios involving noisy or low-quality data, which are common outside controlled benchmark environments? It is crucial to understand the model's robustness and performance in less ideal, real-world conditions to evaluate its practical applicability.

**Comparison with State-of-the-Art Models**

- How does the model's performance compare with state-of-the-art commercial models such as GPT-4v, GPT-4o, and Gemini? Understanding the discrepancies and similarities in performance between this model and leading commercial models would provide a clearer picture of its competitive standing.

**Architectural and Training Insights**

- What is the performance impact of using cross-attention while unfreezing all parameters?
- Can a column for the number of optimized parameters be added in Table 3 to highlight the differences? Including this information would help to better illustrate the comparative efficiency and complexity of the models.
- Does the final model use both LoRA and DoRA? As discussed in the exploration, a full autoregressive structure is better when equipped with LoRA finetuning on the unimodal backbone. Also, the paper mentioned that DoRA is used for instruction finetuning. So, does the final model use both LoRA and DoRA for different purposes during training?

**Model Design and Inference Capacity**

- How many images can the model support during inference?

Minor Issues: Missing citation detail in line 22

**Limitations:**

Generally, this is a good paper with several noticeable limitations:

- The training resources are quite large and may not be accessible to many researchers, limiting the reproducibility of the results.
- The evaluation benchmark is limited. More results on commercial SOTA VLMs like GPT-4v, and on more tasks such as reasoning, hallucination, and in-context learning, would be useful to fully assess the model's performance.
- The red-teaming procedure could be more thorough to prevent malicious use of the proposed model.

---

> ### Author Rebuttal · Authors · 2024-08-03
>
> Thank you for your detailed review. We appreciate your feedback and will address each point individually. We unfortunately cannot cite your questions without exceeding the maximum length.
>
> 1) It is true that our comparisons primarily involve open-source models. This is because state-of-the-art commercial models tend to be significantly larger, making direct comparisons challenging. However, we did include MM1, which is not an open-source model. Additionally, in Table 15 of the Appendix, we provide comparisons with Gemini and Claude 3. While our model approaches their performance on benchmarks like MathVista and TextVQA, it is clear that larger scales are needed to contain extensive factual knowledge in the model’s weights, and perform well on benchmarks like MMMU.
>
> 2) You are raising a good point that this is a very active field, with many concurrent works, and there are surely potential areas for improvement in the proposed model. We compared ourselves to the best models we were aware of at the time of the submission. Besides InternLM-XComposer2-VL known after, we are not aware of any other model having potentially better scores for the same size at the time of the submission.\
> Now, if we had to justify our work against InternLM, we would say that the scope of the study is different: when building our model, one major point was the efficiency at inference time. This involved using a perceiver resampler to reduce the number of visual tokens. While this approach works great to obtain even better performance on most tasks, a high number of tokens is still needed for OCR heavy tasks. We also do not have a lower score on the most tracked benchmark MMMU. Second, we only use open datasets, while they use proprietary data. Finally, we focused on learning from ablations for the researchers/community rather than just giving a model.
>
> 3) We evaluated our base model with 8 in-context examples, as we wanted to compare our model to MM1-base-7B, the best base model in its class size at that time, which was only evaluated using 0-shot (which doesn’t really give information since the model cannot know in which format it should format its answer) and 8-shot.\
> We did see an expected improvement when going from 4-shot to 8-shot by 1-2 points.\
> When evaluating the base model, we are very sensitive to the template expected by the benchmark, which was never seen during training. For example, answering “yes it is” while the growth truth is “yes” would be counted as false. Adding more in-context examples boosts the scores, but there is no evidence at first that it’s improving the model’s ability to solve the task. Potentially, it could only help it to better follow the template expected by the benchmark. A solution to discriminate that would be to use a LLM as judge approach, but it would start to be a bit complex and out of the scope of this current study, and also not comparable with any other work, so this is also a reason why we didn’t push the in-context learning evaluations.
>
> 4) We guaranty that all the datasets used for the training are publicly available. Moreover, we are working on cleaning our codebase and documenting it before opening it.
>
> 5) We agree on the importance of testing under real-world conditions. Our model has been publicly demoed and tested by many users, although we cannot share details here without compromising anonymity. Additionally, our model was tested by many users against many other models with real-world images and prompts, and obtained a strong ELO score.
>
> 6) We provide in Table 15 in Appendix a comparison with the Gemini and Claude 3 series of models.
>
> 7) For both the cross-attention and fully autoregressive architectures, unfreezing all parameters during pretraining led to unstable training and eventual loss divergence, motivating Finding 3 in our paper.
>
> 8) Thank you for this suggestion. We have added the number of trainable and total parameters for each model in Table 3.
>
> 9) The entire pretraining and ablations were done with LoRA. After DoRA was published, we tested it as a direct replacement for LoRA and found similar or slightly better performance. We decided to use DoRA for fine-tuning as it did not hurt performance, did not add more parameters, and was proven to be efficient in various cases by the community.
>
> 10) The model can handle an arbitrary number of images as input, with the maximum number limited by the sequence length it was trained with. In our case, the maximum sequence length was 2048, allowing for just over 30 images. Fine-tuning with a longer sequence length could enable inference with more images, such as for videos.
>
> 11) We have corrected this issue.
>
> 12) Besides the comparison to commercial VLMs mentioned earlier, we acknowledge that the model is not evaluated on a large number of benchmarks (6 in Table 15). This is due to the challenge of finding benchmarks that effectively distinguish VLM performance.\
> Some open-ended benchmarks are unreliable as good scores often indicate adherence to expected answer templates rather than true capability. For example, we obtained 81.2 on VQAv2 while Gemini 1.5 Pro scored 73.2. It doesn’t mean that our model is better than Gemini 1.5 Pro. We kept benchmarks like MMMU for its comprehensive categories, MathVista for reasoning and math abilities, TextVQA and DocVQA for OCR and document understanding, and MMBench for general VQA evaluation.
>
> Thank you again for your detailed comments. We welcome further discussion if you have any additional questions.

---

> > ### Comment · Reviewer_vo36 · 2024-08-08
> >
> > Thanks for the detailed feedback. I'm glad the rebuttal resolved my questions and concerns. I agree that many current benchmarks may be inadequate for reflecting visual-centric understanding ability. Understandably, comparing with SOTA commercial models can be unfair due to mismatched model sizes and pre-training datasets.
> >
> > There is no further question from my side and I keep my score unchanged.

---

### Official Review · Reviewer_Gtma · 2024-07-11

**Soundness:** 2
**Presentation:** 3
**Contribution:** 3
**Rating:** 6
**Confidence:** 3

**Summary:**

The compares different methods and strategies involved in training VLMs
- impact on inference efficiency by model architecture (fusion module: cross-attention versus autoregressive) & on training stability by multimodal training procedure
- compare different design choices in a controlled environment and extract experimental findings
- Build a new  8B model XXXXXX and a new instruction training dataset YYYYYY that surpasses or closely-matches the performance of many large-scale models

**Strengths:**

- The paper has an interesting scientific-report like approach to find a good recipe for training VLMs which would be beneficial to the community
- Using the derived findings, the proposed model XXXXXX performs higher scores than larger models
- The authors also curate a new vision-language finetuning dataset YYYYYY.
- The authors consider many important, yet overlooked design choices, which would help expedite research and development of VLMs

**Weaknesses:**

- While I fully appreciate the importance of the work in standardizing a recipe for training VLMs, I believe it lacks vigour to make such bold claims. For example, Finding 4 does contradict results in Table 9, TextVQA.
- Some results seem a bit too overarching. For example, while the authors make a generalised finding “the quality of the language model backbone has a higher impact on the performance of the final VLM than the quality of the vision backbone” (Finding 1), they do mention that this might be because the vision encoders might not be well trained (L108), which seems conflicting.
- It would be nice to have most/all the findings validated for XXXXXX.
- Can the authors elaborate on how YYYYYY is different from existing VLM instruction tuning datasets? Is there any scope of test dataset leaks?
- How do the authors ensure that the superior performance of YYYYYY is not driven by the dataset alone and only because of their findings?

**Questions:**

see weaknesses

**Limitations:**

Discussed in section A.4

---

> ### Author Rebuttal · Authors · 2024-08-03
>
> Thank you reviewer Gtma for your interesting remarks, we will try to cover them one by one.
>
> > While I fully appreciate the importance of the work in standardizing a recipe for training VLMs, I believe it lacks vigour to make such bold claims. For example, Finding 4 does contradict results in Table 9, TextVQA.
>
> You raise a valid point.\
> Our ablation results show that for most tasks, particularly non-OCR tasks, the number of tokens to encode the image is not critically important.\
> This is significant for applications like robotics, which do not necessarily require strong OCR capabilities but need efficient inference.\
> This is supported by Tables 9 and 15, which show the image splitting strategy's utility primarily for OCR tasks.\
> This finding aligns with other works published on Arxiv after our submission.\
> We agree that Finding 4 should be stated more cautiously. We have rephrased it to: “Reducing the number of visual tokens with learned pooling significantly improves compute efficiency at training and inference while improving performance on non-OCR downstream tasks.”
>
> > Some results seem a bit too overarching. For example, while the authors make a generalised finding “the quality of the language model backbone has a higher impact on the performance of the final VLM than the quality of the vision backbone” (Finding 1), they do mention that this might be because the vision encoders might not be well trained (L108), which seems conflicting.
>
> We understand that Finding 1 might have been confusing.\
> Our intent was to highlight that, given a fixed size for the vision encoder (around 400M for CLIP-H or SigLIP) and a fixed size for the language model (around 7B for Llama1 and Mistral), using the best available open-source models at the time, the most significant performance improvement was observed when upgrading the LLM.\
> We have clarified this by rephrasing the finding with: “Better pre-trained LLMs and vision encoders lead to better performance on multimodal tasks. However, with the best current models for both, the LLM has a higher impact.”
>
> > It would be nice to have most/all the findings validated for XXXXXX.
>
> All the findings are validated for the model XXXXXX, except for Finding 1 which has been done for the cross-attention architecture. However, we have validated that it holds also for the fully-autoregressive architecture as XXXXXX. We provide a detailed explanation for this question specifically in the answer to reviewer FVDr.
>
> > Can the authors elaborate on how YYYYYY is different from existing VLM instruction tuning datasets? Is there any scope of test dataset leaks?
>
> At the time of submission, VLM instruction tuning datasets were primarily of two types: synthetically generated Q/A pairs, mostly using ChatGPT (like Llava), and compilations of existing training sets from different benchmarks (like InstructBLIP).\
> We adopted the second approach, conducting an extensive literature review to compile a mixture of the 50 highest quality datasets we found.\
> To our knowledge, this scale and diversity had not been done before, with many interesting datasets previously missed.\
> We transformed all datasets into a consistent format, created images for benchmarks without them (like tables), merged Q/A pairs on the same images to create multi-turn dialogues, prepared non-prompted datasets with prompts, and removed images present in test sets of benchmarks we evaluate on. We have done a contamination analysis by checking that images present in the splits of the benchmarks we evaluate on are not present in our dataset. We guarantee that the dataset YYYYYY is released for the community.
>
> > How do the authors ensure that the superior performance of YYYYYY is not driven by the dataset alone and only because of their findings?
>
> The dataset YYYYYY certainly plays a crucial role in the final performance, but it is introduced during the fine-tuning stage. The findings are primarily derived before this stage, with the exception of Finding 6, which is independent of the data. Thus, the ablations in Section 3 informed better model and training method choices, while the dataset YYYYYY provided an additional, independent boost.
>
> Thank you again for your review, and we will be happy for a further discussion if you have any additional questions.

---

> > ### Comment · Reviewer_Gtma · 2024-08-08
> >
> > Dear authors,
> >
> > I have read through you responses to my queries -- thank you! I am glad that the authors revised the findings to be less liable to misinterpretation. Hence, I am glad to increase my score from 5 to 6.
> >
> > Also, I agree with other reviewers that the model could benefit from more rigorous evaluation (for example, reporting fine-grained performance on MMBench).

---

> > > ### Author Response · Authors · 2024-08-08
> > > **Answer to reviewer Gtma**
> > >
> > > Thank you for this suggestion, we will then also report in the final version the fine-grained performance on MMMU (rather than MMBench, if possible) given the high number of diverse categories it contains.

---

### Official Review · Reviewer_Ww8w · 2024-07-12

**Soundness:** 3
**Presentation:** 2
**Contribution:** 3
**Rating:** 6
**Confidence:** 4

**Summary:**

This work studies a question: what matters when building vision-language models? To this end, this work provides analysis from the following aspects, 1) Are all pre-trained backbones equivalent for VLMs? 2) How does the fully autoregressive architecture compare to the cross-attention architecture? 3) Where are the efficiency gains? (Number of visual tokens and Preserving the original aspect ratio and image resolution) 4) How can one trade compute for performance? Based on these observations, this work further proposes a vision-language foundation model.

**Strengths:**

- First of all, I like the idea of analyzing the impact of each module on the VLMs (e.g., LLaVA). This work considers several aspects, such as language model, vision encoder, and input resolution.

- Second, introducing a VLM model and giving the specific training recipe. I think the three main stages are straightforward, including multi-stage pre-training, Instruction fine-tuning, and optimizing for chat scenarios. The last one is not used in other VLMs. I think it is useful. Please comment on does this strategy can help other existing methods such as LLaVA?

**Weaknesses:**

- Some of the findings are already studies in previous works. For example, the impact of a vision encoder has been studied in [a,b,c]. The effect of the language model is studied in [a,d]. The impact of input resolution [a,b,d]. However, I do not see a clear comparison and discussion with these works. Namely, although I think the analysis is interesting, it is not new and does not provide any new insights.

   [a] Prismatic VLMs: Investigating the Design Space of Visually-Conditioned Language Models

   [b] Eyes Wide Shut? Exploring the Visual Shortcomings of Multimodal LLMs

   [c] BRAVE : Broadening the visual encoding of vision-language models

   [d] VILA: On Pre-training for Visual Language Models

- Regarding the improvement or training of a new VLM model. While the training recipe is provided, it is crucial to compare it with other methods that also aim to improve VLMs. For example, [b,c,d] all improve the VLMs. I would suggest the authors give a direct comparison or a clear discussion on how the proposed method contributes and why it is important.

**Questions:**

- I understand YYYYYY and XXXXX are set for anonymization. But they indeed affect the reading. [Just a feedback]

- Please discuss the observations/findings in the context of existing works. Moreover, another work which is released after NeurIPS would be useful to look at [d]

    [d] Cambrian-1: A Fully Open, Vision-Centric Exploration of Multimodal LLMs

- Please discuss or compare with methods that also aim to improve VLMs.

**Limitations:**

This work provides a discussion of the potential limitations in Section A.  I agree with the common issue of hallucinations. Could you provide some suggestions on how to avoid them? Second, I do not quite get "lack of openly available". Are the datasets used in this work not publicly available? [I did not force the authors to release the data but a clarification would be helpful]. Lastly, it would be better if the authors could provide some advice on handling "jailbreak" cases.

---

> ### Author Rebuttal · Authors · 2024-08-04
>
> Thank you for your remarks that we’ll address one by one, unfortunately briefly and without citing your questions not to exceed the max length.
>
> 1) Optimization for chat scenarios depends on training data.\
> We used YYYYYY, a compilation of 50 high-quality, mostly human-annotated datasets from literature.\
> These datasets' short Q/A pairs can lead to brief model responses.\
> To address this, we fine-tuned first on YYYYYY for improved performance (reasoning, OCR, …), then briefly fine-tuned on long conversations to adapt output length.\
> LLaVA, trained directly on long conversations, doesn’t need this step.\
> We avoided using current conversational datasets due to their synthetic nature (often ChatGPT-generated) and simplistic, not diverse questions. YYYYYY's creation and open-source release aims to improve VLMs by leveraging diverse, accurate datasets from literature.
>
> 2) We acknowledge the rapid evolution of multimodal vision-language models, with some overlap in findings expected given typical 6-12 month project timelines.
>
> Regarding the very recent papers you mentioned:\
> -[c] BRAVE (April 10): Considered concurrent work per NeurIPS guidelines, given its date.\
> -[a] Prismatic VLMs: We’ve cited it extensively. While there are similarities, we've expanded upon their findings. For instance, while they find that unfreezing the vision encoder degrades significantly the performance, we demonstrated that a LoRA strategy can achieve stable training and improved performance when adding trainable weights to the vision encoder, as mentioned just before Finding 2 in our paper. We also highlight additional differences in section 4.1.\
> -[d] VILA: Also cited, VILA's scope on image resolution was more limited (224 to 336 pixels), while we extend it to 980 pixels and focused more on image splitting strategies rather than comparing the input resolutions for a fixed number of tokens.\
> -[b] "Eyes Wide Shut?": While this paper analyzed the impact of pre-trained vision encoders, it represents only a very small portion of our broader study.
>
> While we appreciate the similarities you've noted (from our contributions from section 3.1), we believe our work offers several unique and significant contributions, particularly in sections 3.2, 3.3, 3.4, and 4:\
> **Most important part for this answer**, please read our general message "Author Rebuttal" to all reviewers for a list of our unique contributions, as we cannot put them here due to length constraints.
>
> These contributions offer new insights to the field beyond the points initially highlighted in your review, that can be now seen as complementary points in our broader study.\
> We've put effort to be comprehensive in our literature review and comparisons against other works throughout the paper, as shown by our extensive bibliography.
>
> We hope this clarification helps to illustrate our contributions. We will do our best to highlight even more these differences in our final version.
>
> 3) We agree that [b,c,d] all contribute to advancing VLMs in various ways. Our work, while complementary to these efforts, introduces several novel approaches and improvements that we believe are unique and significant (cf previous answer).\
> We have provided comparisons with other models throughout our paper. For instance, a key aspect in our strategy is efficiency at inference. We have managed to do so with three points:\
> -Using an efficient pooling strategy with a very small number of visual tokens. This is compared against MM1, SPHINX-2K, DeepSeek-VL, DePALM (some of the strongest VLMs at that time, with better performance than VILA or Prismatic VLMs).\
> -Using the image aspect ratio preserving strategy. To our knowledge, there is no other VLM using this strategy, so it could be compared against all the other models.\
> -Fine-tuning with variable visual token counts (with/without image splitting). Unlike models we compare to like Llava, MM1, and Monkey that fine-tune only with image splitting, our approach allows users to choose token count at inference. This flexibility wasn't the norm before.
>
> 4) Our findings remain relevant in the context of recent works. The field has shifted towards unfreezing language models when feasible or using LoRA for improved stability, aligning with our proposed methods.\
> Most recent works also use image splitting with variable sub-image counts during training, enabling compute-performance trade-offs at inference (section 3.4).\
> Our dataset YYYYYY demonstrated the potential of leveraging diverse existing academic datasets for instruction fine-tuning, previously implemented at a smaller scale with less diversity.\
> The dataset Cambrian-10M is very inspired by our work, containing most of the public academic datasets we have selected, and the images of the tables we have rendered (we created these table images from datasets that originally lacked visual representations, employing various styles to enhance diversity).\
> Notably, our XXXXXX-8B model outperforms Cambrian-1-8B on challenging benchmarks like MMMU and MathVista, while being significantly faster at inference due to fewer visual tokens (64 vs. 576).
>
> 5) To mitigate hallucinations, we suggest: (1) improving image-text alignment in data, (2) using DPO to reduce hallucination probabilities, and (3) incorporating negative prompt datasets.\
> We've released all datasets used. Regarding "lack of openly available", prior works often used LLMs to create synthetic Q/A pairs, limiting diversity and encouraging hallucinations. Our approach involved compiling and processing/modifying diverse, high-quality publicly available datasets.\
> RLHF methods could enhance VLM robustness against jailbreaking.
>
> We hope that our responses have provided a clearer picture of our research's unique aspects and its position within the current landscape of vision-language models.\
> We have tried to show why our work could be impactful and how it's different from what has been done before.

---

> > ### Comment · Reviewer_Ww8w · 2024-08-11
> > **Thank You**
> >
> > Dear Authors,
> >
> > Thank you for your detailed response, which has effectively addressed the initial concerns. Please include a paragraph in the revision that highlights the key observations and insights from the existing works.
> >
> > Best,
> >
> > Reviewer Ww8w

---

> > > ### Author Response · Authors · 2024-08-11
> > > **Answer to reviewer**
> > >
> > > Thank you for your comment, we will add this paragraph in the final version.

---

> > > > ### Comment · Reviewer_Ww8w · 2024-08-12
> > > >
> > > > Dear Authors,
> > > >
> > > > I have re-read the submission and would like to increase the score to 6. Please create a project page to host the models, datasets, and training code.
> > > >
> > > > Best regards,
> > > >
> > > > Reviewer Ww8w

---

> ### Author Response · Authors · 2024-08-12
> **Answer to reviewer**
>
> Thank you for your re-evaluation.
>
> We created a public collection with
> - the datasets used (pre-training + fine-tuning)
> - the models (base, instruct, instruct-chatty)
> - 4-bit quantized versions of the models
> - tutorial to fine-tune the models
>
> easily downloadable.
>
> We are working on the remaining parts.

---

### Author Rebuttal · Authors · 2024-08-04

Dear reviewers, thank you for your detailed remarks. We have commented on each of your point individually.\
Before that, we would like to highlight a summary of our biggest contributions with this work.

- We demonstrate the effectiveness of LoRA training during pre-training for stable training and improved performance (to our knowledge, it wasn’t done before).
- We provide the first comprehensive comparison between cross-attention and fully autoregressive architectures (to our knowledge, it wasn’t done before).
- We show that a learned pooling strategy can significantly improve efficiency at inference while maintaining or improving performance, especially for non-OCR tasks – a finding with important implications for applications like robotics where performing fast inferences is needed (to our knowledge, it wasn’t done with a number of visual tokens as low as 64 before).
- We use the image aspect ratio preserving strategy for arbitrary resolution inputs, improving inference efficiency when lower image resolutions are passed as inputs (to our knowledge, it is not done in any current VLMs).
- We demonstrate that increased token counts for image encoding are primarily beneficial for OCR tasks (to our knowledge, it wasn’t done before).
- We contribute a large-scale, open-source instruction dataset compiled from existing literature (to our knowledge, it wasn’t done at this scale and with this diversity before).
- We provide the community with a strong and efficient 8B open-source VLM (state-of-the-art at the time of the submission).

Other complementary contributions, less “unique”, include:
- A study of the impact of the pre-trained language model and vision encoder on the final VLM.
- The gain obtained by using fully synthetic captions for image-text pairs during the pre-training, which are less noisy than the original alt-texts present on the internet, usually of very poor quality.
- The introduction of the task of text transcription of PDFs directly during the pre-training to increase OCR performance (to our knowledge, it wasn’t done before).

---

### Decision · Program_Chairs · 2024-09-25

**Decision:**

Accept (poster)

**Comment:**

Reviewers and I applaud the in-depth and module-wise analysis of VLM as presented in this draft. Some concrete suggestions made by fellow reviwers and they would improve the draft. The topic is of great interest to the NeurIPS audience. Accept recommended.